# Host Plant Modulated Physio-Biochemical Process Enhances Adaptive Response of Sandalwood (*Santalum album* L.) under Salinity Stress

**DOI:** 10.3390/plants13081162

**Published:** 2024-04-22

**Authors:** Kamlesh Verma, Ashwani Kumar, Raj Kumar, Naresh Kumar, Arvind Kumar, Ajay Kumar Bhardwaj, Ramesh Chander Verma, Prashant Sharma

**Affiliations:** 1ICAR—Central Soil Salinity Research Institute, Karnal 132001, Haryana, India; kamlesh.ugf@gmail.com (K.V.); soninaresh2809@gmail.com (N.K.); arvind.kumar2@icar.gov.in (A.K.); ak.bhardwaj@icar.gov.in (A.K.B.); 2Department of Forestry, CCS Haryana Agricultural University, Hisar 125004, Haryana, India; rcverma8268@gmail.com; 3Department of Silviculture and Agroforestry, Dr. Yashwant Singh Parmar University of Horticulture and Forestry, Solan 173230, Himachal Pradesh, India; prashantsharma92749@gmail.com

**Keywords:** sandalwood, salinity, water relations, antioxidative enzymes, host species

## Abstract

Salinity is one of the most significant abiotic stress that affects the growth and development of high-value tree species, including sandalwood, which can also be managed effectively on saline soils with the help of suitable host species. Therefore, the current investigation was conducted to understand the physiological processes and antioxidant mechanisms in sandalwood along the different salinity gradients to explore the host species that could support sandalwood growth in salt-affected agro-ecosystems. Sandalwood seedlings were grown with ten diverse host species with saline water irrigation gradients (EC_iw_~3, 6, and 9 dS m^−1^) and control (EC_iw_~0.82 dS m^−1^). Experimental findings indicate a decline in the chlorophyll content (13–33%), relative water content (3–23%), photosynthetic (27–61%) and transpiration rate (23–66%), water and osmotic potential (up to 137%), and ion dynamics (up to 61%) with increasing salinity levels. Conversely, the carotenoid content (23–43%), antioxidant activity (up to 285%), and membrane injury (82–205%) were enhanced with increasing salinity stress. Specifically, among the hosts, *Dalbergia sissoo* and *Melia dubia* showed a minimum reduction in chlorophyll content, relative water content, and plant water relation and gas exchange parameters of sandalwood plants. Surprisingly, most of the host tree species maintained K^+^/Na^+^ of sandalwood up to moderate water salinity of EC_iw_~6 dS m^−1^; however, a further increase in water salinity decreased the K^+^/Na^+^ ratio of sandalwood by many-fold. Salinity stress also enhanced the antioxidative enzyme activity, although the maximum increase was noted with host plants *M. dubia*, followed by *D. sissoo* and *Azadirachta indica*. Overall, the investigation concluded that sandalwood with the host *D. sissoo* can be successfully grown in nurseries using saline irrigation water and, with the host *M. dubia,* it can be grown using good quality irrigation water.

## 1. Introduction

Secondary salinization is an emerging environmental problem that degrades land and impedes crucial ecosystem services such as limiting agricultural productivity, hydrological resources, loss of biodiversity, and nutrient recycling [1,2]. In changing climatic scenarios, globally, plants frequently interact with drought and/or salinity stress by decreasing the subterranean water table [3]. Notably, the soils in hot arid regions are primarily saline, and the crops grown in these areas exacerbate the problem of inadequate irrigation management practices, leading to secondary salinization. Salt-affected lands in India encompass an area of approximately 6.74 million hectares (m ha) and pose a significant challenge to the nation’s capacity to enhance food production to meet the growing demand. Moreover, the latest scientific predictions indicate that ~16.2 m ha of land will be salt-affected in the next three decades due to faulty agricultural management practices and climate change [4]. Due to salt-affected soils, India loses 16.84 million tons of agricultural production of a value of ~2.9 billion USD annually [5]. A substantial fraction (32–84%) of groundwater in the semi-arid and arid parts of India is of low quality [6], and its uncontrolled utilization poses a severe threat to the long-term viability of natural resources and the ecosystem. Overall, salinity causes soil degradation, thereby making fertile lands unproductive and resulting in low returns from agriculture [7]. To protect farmers from price/return-related distress, a permanent and long-term solution is urgently required [8].

One of the alternatives is to adopt low-input–high-valued crop (like Sandalwood)-based agroforestry systems to enhance the income and sustainable livelihoods of marginal farmers residing in degraded ecosystems. Sandalwood (*Santalum album* L.), one of the most expensive woods in the world allied with Indian culture, unfortunately still lacks expansion over larger areas due to undesired security threats and a government ban on sandalwood plantations [9,10]. It is renowned for its aromatic oil (East Indian Sandalwood tree oil) obtained from heartwood, which is widely utilized in various industries such as perfumery, medicinal and aromatic industries, incense, therapy, and skin cancer prevention [11,12]. However, a few years ago, the Indian government eliminated all limitations on sandalwood cultivation, and the Planning Commission of India also emphasized the promotion and cultivation of sandalwood on the farmland. Therefore, considering the economic significance of sandalwood, farmers and stakeholders throughout India are expressing tremendous interest [9,13]. Conversely, however, techniques for establishing sandalwood plantations and the recommended set of practices are still being developed [14]. Initially, it was mostly confined to the forests of Southern India, but due to its remarkable adaptability in tropical and sub-tropical regions across India, sandalwood cultivation could be promising in non-conventional areas [13].

Being semi-parasitic in nature, sandalwood plants require host plants that have the potential to grow in combination and provide nutrients and water for better growth of sandalwood plants [15]. Further, in nature, more than 300 plant species can serve as host plants for sandalwood tree by providing nutrients and water through a specialized organ called the haustorium, particularly in the initial stages of growth [16,17], since the sandalwood roots lack root hairs [18]. Specifically, the growth of parasitic angiosperms is regulated by host root-derived chemical signals [19] and 70% of seedlings are capable of producing haustoria within thirty days of germination [20]. However, in the absence of host plant, the leaves of sandalwood plants either shed down or become yellow [21]. Sandalwood water and photosynthetic efficiency are primarily determined by how the host plant responds to environmental factors [17]. Thus, the selection of the appropriate host becomes more crucial, since the host species also influences the haustorial growth, the composition of organic acids, sugars, and amino acids in the xylem stream, the carbon assimilation rate, and the chlorophyll content in sandalwood plants [18]. Hence, it is imperative to promptly identify appropriate host plants that can thrive in challenging conditions, like in saline soils. In particular, salinity stress initially reduces water availability due to osmotic stress. Over time, it leads to the accumulation of detrimental ions, particularly Na^+^ and Cl^−^, which hinder growth and physico-biochemical processes and generate reactive oxygen species [22]. Tolerant species exhibit a greater ratio of potassium (K^+^) to sodium (Na^+^) in their tissues, which enables them to grow more and produce a significant amount of biomass even when exposed to high salt levels [23]. Previously, sandalwood was shown to prefer legumes as hosts, likely because of the legumes higher levels of glutamine synthase activity [24]. However, among leguminous plants, *Acacia acuminata*, a resilient plant species, offers several advantages in terms of improved growth and a more favorable K:Ca ratio than less resilient species such as *Allocasuarina huegeliana* [18,25]. In general, three factors—the choice of sandalwood for hosts, the ability of hosts to tolerate salinity, and the regulation of physiological and biochemical processes in sandalwood—likely determine the salinity tolerance and growth of sandalwood in saline environments.

Consequently, there is a dearth of information regarding the detailed mechanisms of the physiological and biochemical processes by which sandalwood adapts to different stresses, including salinity. Such perceived situations have led to substantial research gaps in terms of growth performance and physiological changes in *S. album*, affecting its adoption and promotion in saline soils. Being hemiparasitic in nature, the identification of a host plant for sandalwood could be the first step toward increasing the salt tolerance of *S. album.* Secondly, how the physio-chemical and plant metabolites of *S. album* change under varied levels of salinity stress was assessed. In the current investigation, we have hypothesized that the salinity tolerance of sandalwood could be increased through haustorial connections if compatible and salt-tolerant hosts are considered. Taking these variables into account, the proposed study has been conceived to address the highlighted issues about assessing sandalwood in saline soils with the following specific objectives: (i) identification of a suitable host plant for sandalwood under saline conditions; (ii) assessment of salinity-induced changes in physiology and redox homeostasis mechanisms in sandalwood. The findings of the present study will offer policymakers unique perspectives on the cultivation of this economically significant tree species in saline environments, with the aim of improving farmers’ livelihoods. Additionally, this research will contribute to advancing knowledge within the scientific community by exploring the physiological mechanisms through which sandalwood adapts to salinity.

## 2. Results

### 2.1. Morphological Parameters

The findings from the current investigation indicated that the diameter of the sandalwood reduced significantly as the salinity varied from 8.32 to 30.51% in comparison to the control condition (6.13 mm) (Appendix A). Similarly, the sandalwood biomass also showed a declining trend, ranging from 22.69 to 65.02% compared to the control treatment (108.75 g). Overall, sandalwood grown with the host species *Melia dubia* and *Dalbergia sissoo* was found to have a higher diameter (6.96 mm and 6.87 mm) and accumulate more biomass (107.89 g and 107.35 g) compared to other selected hosts.

### 2.2. Photosynthetic Pigments

A perusal of the data on the content of photosynthetic pigments in sandalwood (Table 1) revealed that among the different salinity levels, a gradual reduction of 13.1–33.3% in chlorophyll content was recorded with salinity in comparison to the control condition. Among different host species, in control conditions, sandalwood plants retained maximum chlorophyll with host *Casuarina equisetifolia* (1.41 mg g^−1^), which was statistically similar to the hosts *Phyllanthus emblica*, *Citrus aurantium*, and *M. dubia*, while the lowest was recorded with host *Punica granatum* (0.42 mg g^−1^). The minimum decrease in chlorophyll content was recorded with hosts *C. aurantium* (13.24%) and *M. dubia* (13.49%) at EC_iw_~6 dS m^−1^, whereas at salinity (EC_iw_~9 dS m^−1^), hosts *M. dubia* (25.40%), *D. sissoo* (29.23%), and *C. aurantium* (29.23%), recorded the lowest reductions. Carotenoid content enhanced with increasing stress from 1.55 to 2.73 mg g^−1^ FW (fresh weight) (Table 1). At control and EC_iw_~3 dS m^−1^, host *C. aurantium* showed the maximum carotenoid content (1.80 and 2.19 mg g^−1^ FW, respectively). At EC_iw_~6 dS m^−1^ salinity stress, sandalwood grown with *Acacia ampliceps* (2.54 mg g^−1^ FW) had the maximum carotenoid content, whereas at EC_iw_~9 dS m^−1^, sandalwood grown with *D. sissoo* had the maximum carotenoid content (3.60 mg g^−1^ FW). The maximum increase in carotenoid content with salinity stress (up to EC_iw_~9 dS m^−1^) was recorded with hosts *D. sissoo* (113.02%), *M. dubia* (102.37%), and *Azadirachta indica* (98.68%). The least enhancement in carotenoid content was recorded with host *P. granatum* (44.23%).

### 2.3. Gas Exchange Parameters

Sandalwood photosynthetic rate showed a reduction with increasing salinity stress from 4.46 µmol m^−2^ s^−1^ (control) to 1.72 µmol m^−2^ s^−1^ (EC_iw_~9 dS m^−1^) (Table 2). The maximum photosynthetic rate in the control condition was recorded with hosts *Leucaena leucocephala* (7.18 µmol m^−2^ s^−1^) and *D. sissoo* (6.34 µmol m^−2^ s^−1^). Sandalwood photosynthetic rate decreased with salinity, but the minimum reduction was recorded with *D. sissoo* (13.56%), *A. indica* (14.62%), and *M. dubia* (15.91%) at EC_iw_~3 dS m^−1^, and with hosts *M. dubia* (23.55%) and *D. sissoo* (23.82%) at EC_iw_~6 dS m^−1^. At higher salinity stress (EC_iw_~9 dS m^−1^), the minimum reduction in the photosynthetic rate was observed with *D. sissoo* (39.12%), *M. dubia* (39.67%), and *A. indica* (44.44%). The transpiration rate in sandalwood also decreased with salinity stress and the type of host (Table 2). The lowest reduction (up to EC_iw_~9 dS m^−1^) was observed with hosts *M. dubia* (30.94%), *D. sissoo* (44.57%), and *C. equisetifolia* (50.81%), and the maximum reduction was recorded with host *Syzygium cumini* (82.58%).

### 2.4. Plant Water Relations

The findings demonstrated that elevated salinity stress resulted in a reduction in water potential (ψ_p_), and osmotic potential (ψ_s_) (Figure 1). The ψ_p_ of sandalwood decreased by 139.35% in high salinity conditions (EC_iw_~9 dS m^−1^) compared to the control condition (−1.40 MPa) (Figure 1a), whereas the ψ_s_ of sandalwood decreased by 137.47% compared to the control (−4.06 MPa) (Figure 1b). Among the different hosts, sandalwood plants also showed variation in ψ_p_ and ψ_s_ with increasing salinity stress and found a minimum decrease in ψ_p_ with hosts *M. dubia* (25.19% and 57.25%) and *D. sissoo* (34.19% and 58.71%) at EC_iw_~3 and 6 dS m^−1^, respectively. Similarly, the minimum decrease in ψ_s_ of sandalwood at EC_iw_~3 dS m^−1^ was observed with *M. dubia* (24.84%) and *A. indica* (26.32%), and with *P. granatum* (73.20%) and *M. dubia* (83.66%) at salinity level EC_iw_~6 dS m^−1^. However, at higher salinity stress (EC_iw_~9 dS m^−1^), the minimum reduction in ψ_p_ and ψ_s_ was recorded with hosts *D. sissoo* (101.29% and 102.21%) and *A. indica* (105.56% and 109.36%). The relative water content (RWC) also decreased in sandalwood leaves from 77.12% in the control to 59.21% at EC_iw_~9 dS m^−1^ (Figure 1c). Among host plants, the lowest decrease was observed with *D. sissoo* from 74.86% in the control to 66.56% at EC_iw_~9 dS m^−1^, whereas host *C. aurantium* showed maximum RWC reduction from 80.15% in the control to 53.45% under salinity stress (EC_iw_~9 dS m^−1^).

### 2.5. Ion Dynamics

The results pertaining to the ion dynamics (nutrient ratio) indicated a 60.73% reduction in the K^+^/Na^+^ ratio due to salinity (EC_iw_~9 dS m^−1^) compared to the control (Table 3). The highest K^+^/Na^+^ ratio in sandalwood under control was recorded with *M. dubia* (2.17) and *D. sissoo* (2.13), while at salinity level EC_iw_~9 dS m^−1^, hosts *D. sissoo* (1.39), *M. dubia* (1.29), and *A. indica* (1.23) maintained the maximum K^+^/Na^+^ ratio. The minimum K^+^/Na^+^ ratio in both control and salinity conditions was recorded with host *S. cumini*. Furthermore, Ca^2+^/Na^+^ in sandalwood decreased from 1.17 (control) to 0.47 (EC_iw_~9 dS m^−1^). The maximum reduction (up to EC_iw_~9 dS m^−1^) was recorded with host *P. granatum* (63.46%), whereas *D. sissoo* showed the lowest reduction (48.18%). Leaf Ca^2+^/Mg^2+^ in sandalwood decreased with an increase in salinity stress (up to EC_iw_~9 dS m^−1^). However, the minimum decrease (up to EC_iw_~9 dS m^−1^) was recorded with *D. sissoo* (3.03–10.83%) and *M. dubia* (4.22–10.26%), and the maximum with *C. cumini* (20.38%).

### 2.6. Membrane Injury and Lipid Peroxidation

The data presented in Figure 2 indicate that salinity stress increased membrane injury (MI) in sandalwood leaves by 82.06%, 158.53%, and 204.81% (Figure 2a), while lipid peroxidation or malondialdehyde (MDA) content enhanced by 7.82%, 16.94%, and 32.82%, respectively, at EC_iw_~3, 6, and 9 dS m^−1^ compared to the control condition (Figure 2b). Among different hosts, sandalwood plants also showed variation in MI and MDA with increasing salinity and found a minimum increase in MI with hosts *A. ampliceps* (23.34% and 116.00%) and *D. sissoo* at EC_iw_~3 and 6 dS m^−1^, respectively. However, with increased salinity stress (EC_iw_~9 dS m^−1^), a minimum increment in MI was recorded with hosts *A. ampliceps* (180.13%) and *M. dubia*. Similarly, the minimum increase in MDA of sandalwood at EC_iw_~3 dS m^−1^ was observed with hosts *M. dubia* (1.74%) and *A. ampliceps* (4.29%), and with hosts *M. dubia* (6.38%) and *D. sissoo* (10.62%) at salinity level EC_iw_~6 dS m^−1^. Under higher salinity stress (EC_iw_~9 dS m^−1^), a minimum increase in MDA was recorded with hosts *D. sissoo* (21.01%) and *A. ampliceps* (25.69%).

### 2.7. Antioxidant Enzyme

The investigation of the present data (Figure 3) revealed an enhancement of enzymatic activity with increasing salinity stress. Ascorbate peroxidase (APX) enzymatic activity increased with salinity (up to EC_iw_~9 dS m^−1^) by 284.93% compared to the control (Figure 3a). The highest enhancement in APX activity was recorded with host *D. sissoo* (388.24%), whereas the least increase of 134.88% was recorded with host *S. cumini*. The activity of the catalase (CAT) enzyme was also enhanced by 51.72% (EC_iw_~3 dS m^−1^), 144.83% (EC_iw_~6 dS m^−1^), and 287.93% (EC_iw_~9 dS m^−1^) under salinity conditions compared to the control condition (Figure 3b). In salinity stress (EC_iw_~9 dS m^−1^), the maximum enhancement in CAT activity was recorded with *D. sissoo* (178.10%), *A. indica* (170.49%), and *M. dubia* (162.32%), whereas *A. ampliceps* (102.04%) showed the lowest increase in CAT activity (Figure 3b). Peroxidase (POX) enzyme activity was enhanced by 95.98% (up to EC_iw_~9 dS m^−1^) compared to the control. The highest enhancement was recorded with host *D. sissoo* (135.52%) and *L. leucocephala* (78.92%) at EC_iw_~6 dS m^−1^ and with host *D. sissoo* (170.39%) and *M. dubia* (134.61%) EC_iw_~9 dS m^−1^ in comparison to the control (Figure 3c). Superoxide dismutase (SOD) enzyme activity also increased with salinity stress gradient from 8.76 in the control to 25.33 in salinity stress (EC_iw_~9 dS m^−1^). At EC_iw_~9 dS m^−1^, the maximum increment was observed with hosts *D. sissoo* (295.29%), *M. dubia* (243.41%), and *A. indica* (232.85%) (Figure 3d). Another important enzyme, glutathione reductase (GR) activity, also increased with increasing salinity by 18.72%, 40.03%, and 64.23% (EC_iw_~3 dS m^−1^); 144.83% (EC_iw_~6 dS m^−1^); and 287.93% (EC_iw_~9 dS m^−1^), respectively, in comparison to the control (Figure 3e). With salinity, the highest increase in GR activity was observed with hosts *L. leucocephala* (248.78%) and *A. ampliceps* (134.29%), whereas the highest GR activity in the control (1.58) and under salinity stress (EC_iw_~9 dS m^−1^) (2.19) was recorded with host *M. dubia.* However, as an exception, the minimum GR activity decreased with salinity (up to EC_iw_~9 dS m^−1^) by 12.28% in comparison to the control with host *P. granatum*.

### 2.8. Host Plant Species Preference under Salinity Stress

To select the modeled physiological and biochemical traits that contribute to sandalwood biomass under higher salinity stress (EC_iw_~9 dS m^−1^), a linear model (stepwise regression approach) was applied. The regression analysis revealed that a total of 11 traits (RWC, Pn, K, NaK ratio, MDA, APX, POX, SOD, GR, ψ_w_, and diameter) in the control, 10 traits (Pn, Chl, MI, SPAD, Na, K, MDA, APX, SOD, and diameter) in low salinity stress (EC_iw_~3 dSm^−1^), 10 traits (Pn, MI, SPAD, Na, K, POX, GR, ψ_w_, diameter, and RWC) in moderate salinity stress (EC_iw_~6 dS m^−1^), and 12 traits (RWC, E, Chl, MI, SPAD, Na, K, MDA, APX, CAT, GR, and diameter) in higher salinity stress (EC_iw_~9 dS m^−1^), with cumulative R^2^ = 0.98, 0.98, 0.99, and 0.99, respectively, could significantly contribute to explaining biomass variation in sandalwood growth. The regression coefficients (βs) of the respective traits were considered as weighted coefficients for the respective traits mean, and the following formulae were derived for estimating the predicted biomass in various environments (Table 4 and Table 5). Further, a ranking for each tested host species was determined using a multiple regression approach in which biomass was considered as the dependent variable. Based on the predicted biomass response in individual environments, it is revealed that *M. dubia* in a normal growing environment and *D. sissoo* under salinity stress had better ranking, suggesting that they would be the most preferential host for sandalwood growth and development (Appendix A). The host plant species such as *P. granatum*, *A. ampliceps*, and *S. cumini* had lower-ranking values, indicating sensitivity to saline stress. In the control and EC_iw_~3 dS m^−1^, host plant *M. dubia* performed as better host followed by *D. sissoo*, whereas at higher salinity levels i.e., EC_iw_~6 dSm^−1^ and EC_iw_~9 dS m^−1^
*D. sissoo* found to be the best host for sandalwood followed by *M. dubia* (Table 5).

## 3. Discussion

### 3.1. Photosynthetic Pigments

Sandalwood, which is a semi-root parasitic plant, develops its own photosynthetic machinery, but the host plant is critically required for its optimal growth during the later stages [26,27]. The decrease in chlorophyll content of sandalwood with increasing salinity is attributed to the ROS (reactive oxygen species)-induced pigment photo-oxidation and chlorophyll degradation. The reduction in chlorophyll content indicates an approach to safeguard the photosynthetic system from harm caused by salt and prevent the excessive production of ROS, particularly under moderate salinity conditions [28]. Simultaneously, the ROS-induced oxidative stress or increased chlorophyllase activity led to the destruction of chloroplast structure and instability of pigment–protein complexes [29,30] along with a reduction in minerals uptake, such as magnesium, which is required for the biosynthesis of the chlorophyll pigments. Generally, the extent of chlorophyll reduction is directly proportional to the degree of salinity. Moreover, this drop in chlorophyll content indicated that the host–plant relationship was insufficient to deliver the needed nitrogen, which could lead to salinity/nutrient stress-induced photo-inhibition or ROS production [31,32]. Numerous studies have been conducted on different plant species to investigate the effects of salts on photosynthesis pigments and consistently reported that salt stress causes a decrease in photosynthetic rates at low salt concentrations and significant damage to chloroplast structures and photosynthetic machinery at high salt concentrations [33,34,35,36]. Particularly, enhanced chlorophyll and carotenoids in sandalwood with hosts *M. dubia*, *D. sissoo*, and *A. indica* under the control, as well as under different salinity levels (EC_iw_ 3, EC_iw_ 6, and 9 dS m^−1^), might be due to higher nitrogen supply from the host and higher chlorophyll synthase activity [37]. Simultaneously, the differential physiological response of sandalwood with distinct host species might be attributed to the host-induced regulations of light utilization efficiency, closure of stomata, and CO_2_ levels in chloroplasts [38,39].

### 3.2. Gas Exchange Parameters

Gas exchange is a vital physiological process that is directly impacted by reduced leaf expansion, diminished chlorophyll concentration, lower nitrogen levels, changed RuBPase activity, impaired photosynthetic machinery, and senescence. The first and foremost sign of the reduced photosynthetic rate is the stress-induced closure of stomata, which inhibits CO_2_ uptake, causes oxidative damage, and results in no assimilation because the stomatal control is primarily influenced by the moisture level of soil rather than the water content of leaves [40]. The possible explanation for the reduced photosynthesis in mild and moderate stresses might be the closure of stomata due to a decrease in light-use efficiency, leading to CO_2_ deficit in chloroplasts, impaired ATP synthesis, and a significant decrease in ψ_p_ and ψ_s_, which contribute to maintaining a favorable water balance [38,41]. Simultaneously, the higher level of saline conditions can impact photosynthesis activity by affecting non-stomatal factors, such as enzyme activity and carotenoids and chlorophyll concentrations [42]. Similarly, decreased transpiration rate under salinity stress may be attributed to the direct impact of salt on the opening and closing of stomata, resulting in reduced turgor pressure in guard cells and decreased intercellular CO_2_ levels. Additionally, the transpiration rate is influenced by feedback inhibition caused by the reduced efficiency of Rubisco, the activity of sink, displacement of vital cations from endo-membrane structure (resulting in variations in permeability), and swelling of the thylakoid membrane; these factors affect its structure and function, as well as enzyme activities and electron transport [43]. Similarly, Wang et al. [44] also reported a decrease in photosynthetic and transpiration rates with increasing salinity in *Amaranthus tricolor* leaves. Moreover, the current investigation indicated that sandalwood exhibited its highest photosynthetic rate when paired with host plants *P. granatum* and *D. sissoo* under normal conditions. Conversely, the smallest decrease in photosynthetic rate was observed when sandalwood was paired with host plants *D. sissoo*, *M. dubia*, and *A. indica* under conditions of increasing saline stress. The greater resource allocation by these hosts led to greater photosynthesis in sandalwood. The transpiration rate of sandalwood was observed to be the highest with *L. leucocephala* in the control conditions, while the lowest reduction was found with *M. dubia* under increasing salinity stress conditions. The enhanced water absorption and translocation by these host plants might have resulted in greater transpiration in sandalwood.

### 3.3. Plant Water Relations

Water is an important regulator of physiological plant metabolism. Important traits like RWC, ψ_s_, and ψ_p_ are used to measure physiological hydration and elucidate the mechanisms by which plants regulate and sustain the hydration of their cells at an ideal level [45]. Nevertheless, elevated salinity concentrations hinder the transfer of water from the soil to plants, hence impacting the ability of roots to conduct water and the overall water content within plant cells [42]. Simultaneously, salinity can cause cellular oxidative damage and interfere with a variety of cellular functions, including photosynthesis, respiration, plasma membrane function, and turgor loss due to the lower water potential [41]. Under stress conditions, sandalwood maintains the water status by either accumulating osmotically active chemicals or establishing direct xylem-to-xylem connections with host plants [46,47,48]. In addition, the lowering of ψ_p_, ψ_s_, and RWC played a protective role by maintaining osmolyte accumulation, stomatal conductance, turgor pressure, thereby safeguarding macromolecules (such as membranes, chloroplast, and proteins) and their structural integrity from damage caused by stress and facilitating sandalwood adaptability [49,50,51]. In the present findings, the decrease in the ψ_p_ and ψ_s_ of sandalwood under salinity stress can be attributed to limited water availability to root systems. In such conditions, the roots are unable to compensate for the water lost through transpiration by reducing the surface area available for water absorption. The current findings are consistent with previous investigations [52,53,54], which reported the reduction in ψ_p_ and ψ_s_ under salinity stress conditions. Moreover, sandalwood exhibited the highest ψp and ψs values when grown with the host plants *D. sissoo* and *P. granatum* under control conditions. Conversely, sandalwood showed the least reduction in ψp and ψs when grown with *M. dubia*, *D. sissoo*, and *A. indica* under varying levels of salinity stress, indicating greater absorption of water from these host plants by sandalwood.

### 3.4. Ion Dynamics

The primary consequence of salinity stress is to restrict the availability of water through osmotic stress, resulting in a decline of growth and development; on the other hand, in the later stages, toxic ion accumulation, particularly Cl^−^ and Na^+^, play a significant role by disrupting the nutritional balance [38]. The higher absorption rate of Na^+^ and Cl^−^ ions in saline conditions causes a considerable reduction in the absorption of other ions such as K^+^, Ca^2+^, and Mg^2+^ (Appendix A), resulting in the deficiency of both elements in plant tissues; this is also supported by previous literature [55,56]. Sandalwood benefitted through haustoria for absorbing water, nutrients, and carbohydrates, but the deleterious effects of salt stress forced host plants to efficiently exchange toxic ions in a bidirectional manner. Na^+^ toxicity is a major factor contributing to cell damage that mainly interferes with water and nutrient transport through the xylem/phloem. Due to the comparable chemical features of Na^+^ and K^+^ (such as ionic radius and ion hydration energy), Na^+^ negatively impacts the absorption of K^+^ by the root, particularly through nonselective cation channels and high-affinity potassium transporters. Furthermore, it vies with K^+^ for prominent binding sites in crucial metabolic processes within the cytoplasm, including ribosome functions, protein synthesis, and enzymatic reactions [23]. Hence, the cytosol upholds a diminished level of Na^+^ or a reduced Na^+^/K^+^ ratio within the cells when exposed to salt stress [57], as revealed in the present study, where sandalwood plants grown with tree host species showed lower Na^+^/K^+^ ranging from 0.55 (*D. sissoo*) to 1.52 (*C. aurantium*). Specifically, the better performance of sandalwood with *D. sissoo* has already been proven by previous researchers. The current findings indicated that higher levels of Na^+^ content result in decreased K^+^ content, suggesting a competition between both ions for the same binding sites and substitution of K^+^ with Na^+^ at the reaction center. Previous investigations have also suggested that salt stress amplifies Na^+^ uptake and accumulation, leading to leakage and efflux of K^+^ ions from the cells of plants [58]. Consequently, when the concentration of Na^+^ exceeds that of K^+^, it leads to inadequate absorption of nutrients and disrupts the balance of Na^+^/K^+^ in the plant tissues [59], thereby adversely affecting the growth of plants [60,61]. Moreover, plants’ ability to endure osmotic stress depends on their capacity to uphold a high Ca^2+^/Na+ ratio and inhibit the influx of Na^+^ ions. The rise in Ca^2+^ had an opposing effect on the accessibility of Mg^2+^ to the plant since it removed Mg^2+^ from the soil complex [62].

### 3.5. Membrane Injury and Lipid Peroxidation

Though the host species support the growth and physiology of sandalwood plants under salt stress, the response was significantly varied with gradient salinity stress. The MDA is produced due to the cell membrane’s peroxidation under stress as induced by the membrane’s increased permeability and decreased stability, which ultimately leads to unsaturated fatty acids being peroxidized by ROS in plant cells [30]. Moreover, salinity stress increased the leaf membrane injury as well as MDA content (byproducts that could reflect the degree of the peroxidation of the membrane lipid) in both sandalwood and host plants. Sandalwood grown with *M. dubia*, *A. indica*, and *D. sissoo* showed lower MI and MDA content at increased salinity levels. This reduction plays an important role in protecting against oxidative damage caused by salinity and likely contributes to the plant’s ability to adapt to salinity stress [38,63]. This phenomenon may also be associated with the heightened activity of antioxidant enzymes that react with reactive oxygen species (ROS), like O_2_^−^ and H_2_O_2_, thereby mitigating the damage caused to the cell membrane, implying the highest level of resistance to salt stress [39,64].

### 3.6. Antioxidant Enzyme

During abiotic stress, the balance between the generation and removal of ROS is disrupted in plants, leading to harmful oxidation of DNA, lipids, and proteins, as well as the rupture of cell membranes and the loss of cellular solutes [65]. Collectively, plants employ many tactics to counteract the negative effects of salt by collecting a range of osmolytes and antioxidant enzymes. These substances serve to maintain the stability of proteins and membranes by preserving the osmotic balance and preventing the generation of harmful ROS [66]. In the present investigation, the enzymatic activities of POX, CAT, SOD, APX, and GR were enhanced with a progressive increase in salinity stress, counterbalancing the detrimental effects generated by the ROS. The increasing saline stress (3 to 9 dS m^−1^) leads to a gradually higher APX (51.7–287.9%), SOD (53.4–189.2%), CAT (41.7–15.0%), POX (26.5–96.0%), and GR (19.1–64.0) activity, compared to the control condition. Similar to our findings, previous studies [57,67,68,69,70] have shown that increased salinity stress increases the activity of antioxidant enzymes in many plant species.

Plants have highly well-developed adaptation systems and measures of protection to various environmental circumstances. As a result of the elevated level of ROS within the cell, oxidative stress implies a disruption of the redox equilibrium or balance between antioxidants and oxidants, favoring the oxidants and resulting in redox signaling disruption [71,72]. The equilibrium between ROS generation and removal by antioxidant substances and enzymes regulates the ROS homeostasis in cells. Nevertheless, the augmentation of ROS-scavenging enzymes does not consistently correlate with enhanced salt tolerance. Several factors influence the efficacy of antioxidant systems, such as the location where antioxygenic enzymes are produced, the functioning of enzymes, and the interplay between various antioxidant enzymes [42]. The host species supports the growth of sandalwood and counteracts the negative impacts of salt stress by inducing better thermostability to membranes and higher osmolyte accumulation with a better antioxidative defense system. The optimum host association with sandalwood leads to a higher increase in these compounds under stress conditions, making a lesser impact of oxidative damage in sandalwood plants under stressful conditions. The comparison of the host-dependent response pattern in sandalwood for SOD, CAT, APX, and POX activities indicated that *M. dubia* followed by *D. sissoo* and *A. indica* maintained a balance between the enzyme activities, and ROS mitigation revealed that these are the efficient hosts for sandalwood adaptation. Overall, these results showed a variable response of sandalwood towards enzymatic defense, and it was found that the sandalwood grown with *M. dubia* and *D. sissoo* exhibited enhanced levels of antioxidative enzyme activity to counterbalance the detrimental impact of salt stress.

## 4. Materials and Methods

### 4.1. Experimental Area and Planting Material

The investigation was initiated in October 2020 at ICAR–CSSRI, Karnal, India (29°84′30″ N and 76°85′80″ E), and terminated in May–June 2021. The area has a semi-arid and subtropical-monsoonal climate with an average of 350–400 mm of annual rainfall, 70–80% of which is received from July to September. The coldest months are December and January, when the lowest temperature may be as low as 0 °C. The summers are extremely hot, and the mean maximum temperature varies from 40 to 45 °C during May and June. The sandalwood seeds were acquired from the IWST, Bangalore, while seeds of 10 host species i.e., *Azadirachta indica*, *Melia dubia*, *Casuarina equisetifolia*, *Punica granatum*, *Acacia ampliceps*, *Citrus aurantium*, *Phyllanthus emblica*, *Dalbergia sissoo*, *Syzygium cumini*, and *Leucaena leucocephala* were procured from the commercial certified nursery (Appendix A). The seeds of host species and sandalwood were first treated with GA_3_ and sown in germination beds at a uniform depth of 2–3 cm (soil and sand) and watered daily till the completion of germination. The germinated seedlings were transferred to plastic pots containing a mixture of soil, sand, and FYM (6:3:1), watered every third day based on potential evapotranspiration, and allowed to grow in shaded net house conditions. The shaded net house has optimum growing conditions of light intensity (PAR) of 600 flux, relative humidity >60%, temperature of 25–30 °C, and CO_2_ concentration of 400 ppm.

### 4.2. Experiment Description

The experiment was carried out using a randomized block design, with each treatment being replicated five times. The uniformly grown sandalwood seedlings (30 cm height) were transplanted with 10 host species into 10 kg plastic pots containing a mixture of soil–sand–FYM (6:3:1). A spacing of 10 cm was consistently maintained between the sandalwood and host plant in order to facilitate the establishment of haustorial connections. The initial status of the soil mixture is given in Appendix A. For the imposition of salinity stress, all 200 pots of sandalwood (10 hosts × 4 salinity level × 5 replications) were watered uniformly using saline water with the required electric conductivity (EC), i.e., EC_iw_~3, 6, and 9, dS m^−1^ along with the control treatment (best available water of EC_iw_~0.82 dS m^−1^). Pots were irrigated with 500 mL water daily from October to November and February to March, and alternately from December to January. The underground saline water (EC_iw_~16 dS m^−1^) was brought from the Nain experimental farm, ICAR-CSSRI, village Nain, Panipat, and stored in a plastic container. The required quantity of good quality water was mixed with the saline water (EC_iw_~16 dS m^−1^) to maintain EC_iw_~3, 6, and 9 dS m^−1^ water for saline irrigation. Alternatively, saline and good quality water were applied to the seedlings to uniformly distribute salts to maintain the required salinity and prevent the excess accumulation of salts. The ionic composition of saline water used is presented in Appendix A. To sustain plant growth, a standard amount of fertilizers (1 g of NPK per plant) was administered, in addition to a hoagland solution for micronutrients.

### 4.3. Physiological Measurement

After 120 days of salinity treatment, various physiological parameters were measured to assess the impact of salinity stress on sandalwood. For the measurement of the greenness of leaves, the chlorophyll (Chl) content (destructive method, [73]) and carotenoid content [74] were determined. The fully developed top five leaves were tagged, and these tagged leaves were further used to measure the photosynthetic rate (Pn) and transpiration rate (E) using the Portable Photosynthesis System (LI-6800, LICOR Inc., Lincoln, NE, USA). Cuvette conditions were kept under specific conditions, including 25 °C leaf temperature, more than 60% relative humidity, 400 ppm ambient CO_2_ concentration, and 1000 µmol m^−2^ s^−1^ photosynthetic photon flux density [75]. The water potential (ψ_w_; −Mpa) and osmotic potential (ψ_s_) were ascertained according to the methodology described by Kaur et al. [76]. For measuring the relative water content (RWC) [77], 8–10 uniform-size pieces of fresh leaves were harvested for measuring fresh weight (*FW*) and immersed in double distilled water for 4 h to note the turgid weight (*TW*). Subsequently, these pieces were subjected to a drying process at a temperature of 70 °C for 72 h or till the attainment of constant dry weight (*DW*). The RWC (%) was determined by applying the following formula:$$RWC(\%)=\frac{FW-DW}{TW-DW}\times 100$$

### 4.4. Ions Dynamics

To ascertain the levels of ions (Na^+^, K^+^, Ca^2+^, and Mg^2+^) in sandalwood leaves, five fully grown leaves of sandalwood per host and salinity treatment were dried in an oven at a temperature of 60 °C until they reached a consistent weight. After drying, a 0.1 g sample was grounded and homogenized into a powder (fine) using a pestle and mortar. To facilitate digestion, a 10 mL solution of a di-acidic combination (HNO_3_:HclO_4_, 3:1, volume ratio) was used, and the ions were then determined using the methodology outlined by Nguyen et al. [78].

### 4.5. Biochemical Traits

The membrane injury (MI) was determined by adopting the methodology outlined by Dionisio-Sese and Tobita [79] and expressed as %, while lipid peroxidation in terms of malondialdehyde (MDA) content was estimated utilizing 2-thiobarbituric acid and trichloroacetic acid [80] and presented as µmol g^−1^ FW. The different antioxidant enzymes namely catalase (CAT), ascorbate peroxidase (APX), superoxide dismutase (SOD), and peroxidase (POX) were determined using the protocol described by Kumar et al. [57]. Glutathione reductase (GR) was assayed by utilizing the methodology ascertained by Halliwell and Foyer [81].

### 4.6. Statistical Analysis

The observations for each variable were examined for their normality distribution using the Spahiro–Wilk test. The experimental data for each parameter of sandalwood grown with ten selected host species was analyzed using a one-way analysis of variance (ANOVA) at four salinity levels separately using STAR statistical software 2.0.1 [82]. This analysis was conducted to determine the most suitable host species for each salinity level. Simultaneously, a two-way analysis of variance (ANOVA) was also conducted to draw the cumulative influence of the host species and salinity level on the sandalwood. Treatment comparisons were performed using Tukey’s HSD test at a 5% significance level with *p* < 0.05 for multiple mean comparisons to estimate the significant differences between the treatment and host effects. Important physio-biochemical traits were prioritized using a stepwise regression approach in STAR statistical software [82].

## 5. Conclusions

The current investigation concluded that the applied saline levels had a minimal to moderate impact on the physiological processes of the sandalwood. However, physiological processes and antioxidant mechanisms in sandalwood were primarily regulated by the host species, both in salt stress and control conditions. Overall, sandalwood has a great potential for cultivation in saline soils with a suitable host, particularly in areas with an EC_iw_ of 9 dS m^−1^, by maintaining photosynthetic and transpiration content, lower production of MI, and MDA content, as well as by maintaining cellular homeostasis through higher K^+^/Na^+^ and better antioxidative defense system. Moreover, our investigation reveals that sandalwood could be successfully grown using saline irrigation water with host species *D. sissoo* and using good quality irrigation water with host species *M. dubia*. The results of this study will provide researchers and policymakers with distinct insights into the cultivation of this economically important tree species in saline environments, with the goal of enhancing the livelihoods of farmers. However, the current investigation is limited to nursery conditions (controlled environment), emphasizing the need for long-term field experiments to strengthen the reliability of the current findings. Furthermore, moving forward, it is important to prioritize the investigation of how host plants have formed the adaptive response in sandalwood, as well as explore other strategies to further increase this adaptive response.

## Figures and Tables

**Figure 1 plants-13-01162-f001:**
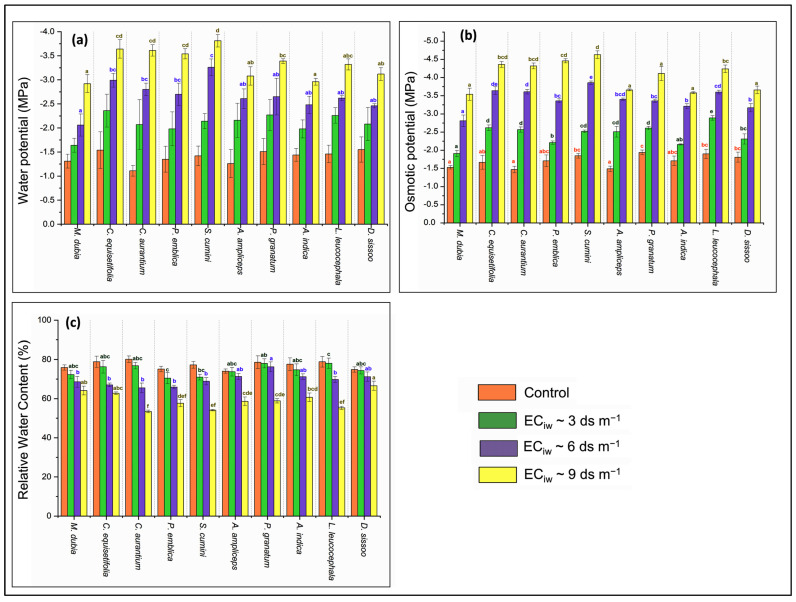
Effect of salinity stress on (**a**) water potential (MPa), (**b**) osmotic potential (MPa), and (**c**) relative water content (%) of sandalwood plants interacting with different host plants (*n* = 5). The values carrying different alphabetical superscripts ^(a–f)^ within the bars above differ significantly among themselves (*p* < 0.05). The error bars indicate standard deviations.

**Figure 2 plants-13-01162-f002:**
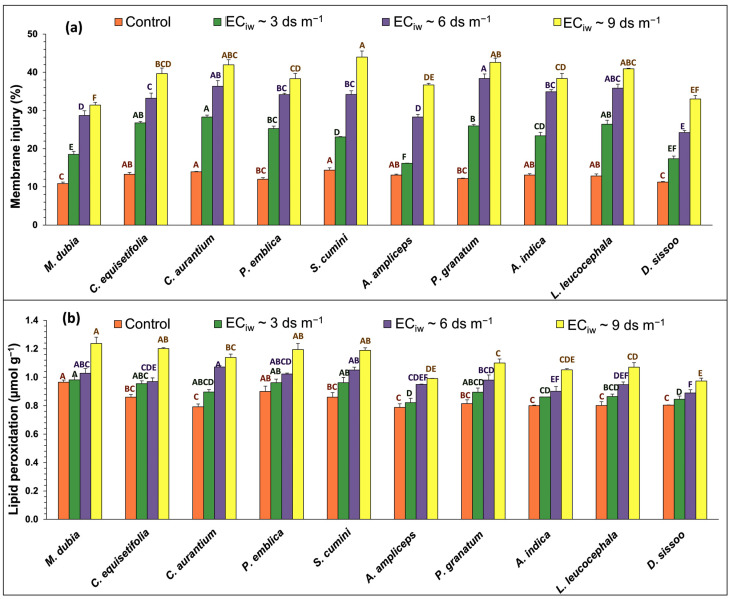
Effect of salinity stress on (**a**) membrane injury and (**b**) lipid peroxidation of sandalwood plants interacting with different host species (*n* = 5). The values carrying different alphabetical superscripts (^A–F^) within the bars above differ significantly among themselves (*p* < 0.05). The error bars indicate standard deviations.

**Figure 3 plants-13-01162-f003:**
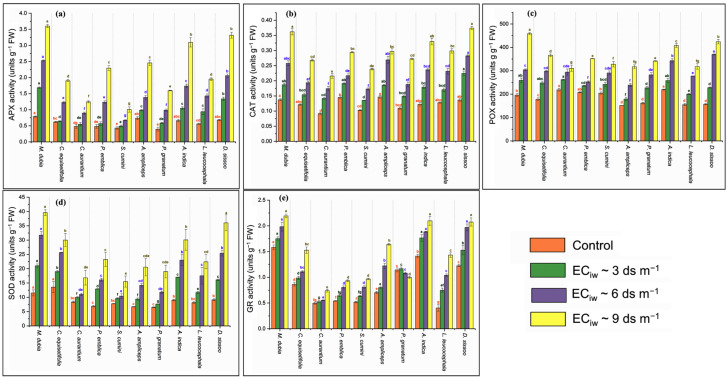
Effect of salinity stress on (**a**) APX, (**b**) CAT, (**c**) POX, (**d**) SOD, and (**e**) GR activity of sandalwood plants interacting with different host plants (*n* = 5). The values carrying different alphabetical superscripts (^a–g^) within the bars above differ significantly among themselves (*p* < 0.05). The error bars indicate standard deviations.

**Table 1 plants-13-01162-t001:** Effect of various host species on the photosynthetic pigment of sandalwood under salinity stress (*n* = 5).

Salinity/ Host Species	Chlorophyll (mg g^−1^)					Carotenoids (mg g^−1^ FW)				
	Control	EC_iw_ 3.0 dS m^−1^	EC_iw_ 6.0 dS m^−1^	EC_iw_ 9.0 dS m^−1^	Mean	Control	EC_iw_ 3.0 dS m^−1^	EC_iw_ 6.0 dS m^−1^	EC_iw_ 9.0 dS m^−1^	Mean
*M. dubia*	1.26 ^AB^ ± 0.09	1.19 ^A^ ± 0.10	1.09 ^BC^ ± 0.04	0.94 ^A^ ± 0.03	1.12 ^b^	1.69 ^A^ ± 0.18	2.17 ^A^ ± 0.23	2.48 ^AB^ ± 0.18	3.42 ^AB^ ± 0.21	2.44 ^a^
*C. equisetifolia*	1.41 ^A^ ± 0.10	1.20 ^A^ ± 0.02	1.07 ^C^ ± 0.02	0.79 ^B^ ± 0.03	1.12 ^b^	1.54 ^A^ ± 0.17	1.84 ^A^ ± 0.18	2.04 ^CD^ ± 0.19	2.63 ^CD^ ± 0.18	2.01 ^cd^
*C. aurantium*	1.36 ^AB^ ± 0.03	1.24 ^A^ ± 0.03	1.18 ^A^ ± 0.03	0.96 ^A^ ± 0.04	1.18 ^a^	1.80 ^A^ ± 0.09	2.19 ^A^ ± 0.16	2.43 ^AB^ ± 0.17	2.98 ^ABCD^ ± 0.19	2.35 ^ab^
*P. emblica*	1.40 ^A^ ± 0.05	1.21 ^A^ ± 0.04	1.15 ^AB^ ± 0.03	1.01 ^A^ ± 0.04	1.19 ^a^	1.48 ^A^ ± 0.33	1.82 ^AB^ ± 0.13	1.99 ^CD^ ± 0.26	2.37 ^D^ ± 0.22	1.91 ^d^
*S. cumini*	1.22 ^B^ ± 0.04	1.02 ^B^ ± 0.04	0.98 ^D^ ± 0.03	0.84 ^B^ ± 0.02	1.01 ^c^	1.50 ^A^ ± 0.21	1.85 ^A^ ± 0.08	2.01 ^CD^ ± 0.10	2.49 ^CD^ ± 0.10	1.96 ^cd^
*A. ampliceps*	0.87 ^C^ ± 0.04	0.73 ^C^ ± 0.03	0.72 ^E^ ± 0.03	0.59 ^C^ ± 0.04	0.73 ^d^	1.77 ^A^ ± 0.22	2.19 ^A^ ± 0.20	2.54 ^A^ ± 0.26	2.90 ^BCD^ ± 0.13	2.35 ^ab^
*P. granatum*	0.42 ^E^ ± 0.01	0.40 ^F^ ± 0.04	0.34 ^H^ ± 0.03	0.30 ^E^ ± 0.01	0.36 ^h^	1.04 ^B^ ± 0.19	1.25 ^B^ ± 0.14	1.29 ^E^ ± 0.21	1.50 ^E^ ± 0.20	1.27 ^e^
*A. indica*	0.71 ^D^ ± 0.03	0.62 ^CD^ ± 0.04	0.59 ^F^ ± 0.02	0.40 ^D^ ± 0.02	0.58 ^e^	1.52 ^A^ ± 0.20	1.88 ^A^ ± 0.34	2.25 ^BC^ ± 0.30	3.02 ^ABC^ ± 0.30	2.17 ^bc^
*L. leucocephala*	0.61 ^D^ ± 0.05	0.46 ^EF^ ± 0.03	0.38 ^H^ ± 0.03	0.32 ^E^ ± 0.01	0.44 ^g^	1.48 ^A^ ± 0.05	1.75 ^AB^ ± 0.14	1.94 ^D^ ± 0.17	2.40 ^CD^ ± 0.18	1.89 ^d^
*D. sissoo*	0.65 ^D^ ± 0.06	0.55 ^DE^ ± 0.02	0.51 ^G^ ± 0.02	0.46 ^D^ ± 0.03	0.54 ^f^	1.69 ^A^ ± 0.19	2.12 ^A^ ± 0.19	2.47 ^AB^ ± 0.22	3.60 ^A^ ± 0.29	2.47 ^a^
Mean	0.99 ^a^	0.86 ^b^	0.80 ^c^	0.66 ^d^		1.55 ^d^	1.91 ^c^	2.14 ^b^	2.73 ^a^	
HSD_0.05_(Host)	0.16	0.13	0.07	0.08	0.03	0.34	0.59	0.28	0.62	0.22
HSD_0.05_(Salinity)	0.04					0.32				
HSD_0.05_(Host × Salinity)	0.07					0.54				

The values carrying different alphabetical superscripts (^A–H^) within the columns above differ significantly among themselves (*p* < 0.05). Treatment means of host and salinity compared with alphabet ^a–h^. Values are mean ± standard deviations.

**Table 2 plants-13-01162-t002:** Effect of various host species on photosynthetic parameters of sandalwood under salinity stress (*n* = 5).

Salinity/ Host Species	Photosynthetic Rate (µ mol m^−2^ s^−1^)					Transpiration Rate (mmol m^−2^ s^−1^)				
	Control	EC_iw_ 3.0 dS m^−1^	EC_iw_ 6.0 dS m^−1^	EC_iw_ 9.0 dS m^−1^	Mean	Control	EC_iw_ 3.0 dS m^−1^	EC_iw_ 6.0 dS m^−1^	EC_iw_ 9.0 dS m^−1^	Mean
*M. dubia*	4.84 ^B^ ± 0.26	4.07 ^B^ ± 0.10	3.70 ^B^ ± 0.30	2.92 ^B^ ± 0.31	3.88 ^c^	1.39 ^E^ ± 0.15	1.21 ^D^ ± 0.12	1.07 ^C^ ± 0.10	0.96 ^C^ ± 0.15	1.16 ^g^
*C. equisetifolia*	3.53 ^C^ ± 0.12	2.75 ^DE^ ± 0.13	1.98 ^D^ ± 0.11	1.43 ^DE^ ± 0.10	2.42 ^g^	1.24 ^E^ ± 0.14	1.07 ^D^ ± 0.11	0.77 ^C^ ± 0.07	0.61 ^E^ ± 0.08	0.92 ^h^
*C. aurantium*	2.59 ^DE^ ± 0.05	1.57 ^F^ ± 0.14	1.17 ^EF^ ± 0.14	0.23 ^F^ ± 0.08	1.39 ^i^	2.27 ^D^ ± 0.12	1.43 ^D^ ± 0.17	1.00 ^C^ ± 0.09	0.53 ^E^ ± 0.05	1.31 ^g^
*P. emblica*	3.09 ^CDE^ ± 0.07	2.48 ^E^ ± 0.11	1.72 ^DE^ ± 0.08	1.05 ^E^ ± 0.18	2.08 ^h^	2.23 ^D^ ± 0.15	1.38 ^D^ ± 0.13	1.05 ^C^ ± 0.10	0.62 ^DE^ ± 0.10	1.32 ^g^
*S. cumini*	2.47 ^E^ ± 0.19	1.08 ^F^ ± 0.08	0.64 ^F^ ± 0.08	0.25 ^F^ ± 0.05	1.11 ^j^	3.33 ^C^ ± 0.17	2.90 ^C^ ± 0.19	1.15 ^C^ ± 0.13	0.58 ^E^ ± 0.07	1.99 ^f^
*A. ampliceps*	4.68 ^B^ ± 0.32	3.66 ^BC^ ± 0.29	2.40 ^CD^ ± 0.13	0.95 ^E^ ± 0.15	2.92 ^e^	6.32 ^A^ ± 0.13	4.88 ^A^ ± 0.16	3.19 ^A^ ± 0.26	1.87 ^B^ ± 0.11	4.06 ^b^
*P. granatum*	6.47 ^A^ ± 0.65	3.20 ^CD^ ± 0.13	2.84 ^C^ ± 0.11	2.01 ^C^ ± 0.17	3.63 ^d^	4.51 ^B^ ± 0.37	3.03 ^C^ ± 0.10	2.27 ^B^ ± 0.15	1.66 ^B^ ± 0.12	2.87 ^d^
*A. indica*	3.42 ± 0.10	2.92 ^DE^ ± 0.11	2.29 ^CD^ ± 0.07	1.90 ^CD^ ± 0.14	2.63 ^f^	3.40 ^C^ ± 0.12	2.84 ^C^ ± 0.20	2.21 ^B^ ± 0.14	0.93 ^CD^ ± 0.05	2.34 ^e^
*L. leucocephala*	7.18 ^A^ ± 0.49	5.20 ^A^ ± 0.24	3.85 ^B^ ± 0.59	2.62 ^B^ ± 0.32	4.71 ^b^	6.69 ^A^ ± 0.11	4.93 ^A^ ± 0.30	3.38 ^A^ ± 0.20	1.97 ^B^ ± 0.15	4.24 ^a^
*D. sissoo*	6.34 ^A^ ± 0.12	5.48 ^A^ ± 0.43	4.83 ^A^ ± 0.12	3.86 ^A^ ± 0.12	5.12 ^a^	4.60 ^B^ ± 0.22	3.88 ^B^ ± 0.11	3.29 ^A^ ± 0.12	2.55 ^A^ ± 0.13	3.58 ^c^
Mean	4.46 ^a^	3.24 ^b^	2.54 ^c^	1.72 ^d^		3.60 ^a^	2.76 ^b^	1.94 ^c^	1.23 ^d^	
HSD_0.05_(Host)	0.845	0.615	0.700	0.541	0.07	0.54	0.48	0.39	0.31	0.16
HSD_0.05_(Salinity)	0.14					0.12				
HSD_0.05_(Host × Salinity)	0.28					0.41				

The values carrying different alphabetical superscripts (^A–F^) within the columns above differ significantly among themselves (*p* < 0.05). Treatment means of host and salinity compared with alphabet ^a–j^. Values are mean ± standard deviations.

**Table 3 plants-13-01162-t003:** Effect of various host species on a nutrient ratio of sandalwood under salinity stress (*n* = 5).

Alinity/ Host Species	K^+^/Na^+^					Ca^2+^/Na^+^					Ca^2+^/Mg^2+^				
	Control	EC_iw_ 3.0 dS m^−1^	EC_iw_ 6.0 dS m^−1^	EC_iw_ 9.0 dS m^−1^	Mean	Control	EC_iw_ 3.0 dS m^−1^	EC_iw_ 6.0 dS m^−1^	EC_iw_ 9.0 dS m^−1^	Mean	Control	EC_iw_ 3.0 dS m^−1^	EC_iw_ 6.0 dS m^−1^	EC_iw_ 9.0 dS m^−1^	Mean
*M. dubia*	2.17 ^A^ ± 0.09	1.72 ^A^ ± 0.03	1.29 ^AB^ ± 0.02	0.89 ^B^ ± 0.02	1.52 ^a^	1.47 ^A^ ± 0.062	1.15 ^A^ ± 0.007	0.86 ^AB^ ± 0.008	0.66 ^B^ ± 0.019	1.04 ^a^	1.95 ^ABC^ ± 0.03	1.87 ^AB^ ± 0.12	1.80 ^A^ ± 0.02	1.75 ^AB^ ± 0.11	1.84 ^bc^
*C. equisetifolia*	1.88 ^BCD^ ± 0.07	1.43 ^BC^ ± 0.06	1.09 ^C^ ± 0.06	0.72 ^CD^ ± 0.04	1.28 ^c^	1.28 ^B^ ± 0.009	0.96 ^B^ ± 0.069	0.73 ^C^ ± 0.009	0.51 ^D^ ± 0.009	0.87 ^c^	2.12 ^AB^ ± 0.09	1.99 ^A^ ± 0.12	1.85 ^A^ ± 0.06	1.72 ^AB^ ± 0.05	1.92 ^ab^
*C. aurantium*	1.79 ^CD^ ± 0.06	1.37 ^BCD^ ± 0.02	1.01 ^CD^ ± 0.05	0.64 ^E^ ± 0.01	1.20 ^e^	1.09 ^CD^ ± 0.030	0.79 ^CD^ ± 0.001	0.60 ^DE^ ± 0.017	0.40 ^EF^ ± 0.012	0.72 ^d^	1.99 ^ABC^ ± 0.04	1.86 ^AB^ ± 0.01	1.72 ^AB^ ± 0.08	1.60 ^BC^ ± 0.08	1.79 ^c^
*P. emblica*	1.82 ^CD^ ± 0.01	1.38 ^BCD^ ± 0.07	1.00 ^CD^ ± 0.02	0.65 ^E^ ± 0.03	1.21 ^de^	0.98 ^DE^ ± 0.036	0.71 ^D^ ± 0.034	0.53 ^EF^ ± 0.031	0.36 ^G^ ± 0.014	0.64 ^ef^	2.05 ^AB^ ± 0.13	1.95 ^A^ ± 0.09	1.78 ^A^ ± 0.05	1.66 ^AB^ ± 0.06	1.86 ^abc^
*S. cumini*	1.67 ^D^ ± 0.13	1.22 ^D^ ± 0.08	0.91 ^DE^ ± 0.01	0.63 ^E^ ± 0.01	1.11 ^f^	0.99 ^DE^ ± 0.055	0.72 ^D^ ± 0.044	0.51 ^F^ ± 0.010	0.37 ^FG^ ± 0.015	0.65 ^ef^	2.00 ^ABC^ ± 0.06	1.89 ^A^ ± 0.06	1.73 ^AB^ ± 0.08	1.59 ^BC^ ± 0.03	1.80 ^c^
*A. ampliceps*	1.90 ^BC^ ± 0.04	1.46 ^B^ ± 0.03	1.03 ^C^ ± 0.03	0.76 ^C^ ± 0.02	1.29 ^c^	0.94 ^E^ ± 0.037	0.69 ^D^ ± 0.034	0.48 ^F^ ± 0.009	0.35 ^G^ ± 0.018	0.62 ^f^	1.79 ^C^ ± 0.06	1.67 ^B^ ± 0.09	1.54 ^B^ ± 0.05	1.44 ^C^ ± 0.02	1.61 ^d^
*P. granatum*	1.76 ^CD^ ± 0.08	1.29 ^CD^ ± 0.03	0.87 ^E^ ± 0.03	0.66 ^DE^ ± 0.03	1.14 ^ef^	1.04 ^CDE^ ± 0.054	0.76 ^CD^ ± 0.035	0.53 ^EF^ ± 0.023	0.38 ^FG^ ± 0.020	0.68 ^e^	2.01 ^ABC^ ± 0.09	1.88 ^A^ ± 0.06	1.75 ^AB^ ± 0.03	1.62 ^BC^ ± 0.02	1.81 ^c^
*A. indica*	2.10 ^AB^ ± 0.10	1.67 ^A^ ± 0.07	1.23 ^B^ ± 0.03	0.86 ^B^ ± 0.01	1.47 ^b^	1.36 ^AB^ ± 0.069	1.05 ^AB^ ± 0.019	0.79 ^BC^ ± 0.045	0.57 ^C^ ± 0.002	0.94 ^b^	2.07 ^AB^ ± 0.04	1.98 ^A^ ± 0.08	1.91 ^A^ ± 0.10	1.81 ^A^ ± 0.02	1.94
*L. leucocephala*	1.86 ^CD^ ± 0.01	1.39 ^BC^ ± 0.04	1.10 ^C^ ± 0.05	0.73 ^C^ ± 0.02	1.27 ^cd^	1.13 ^C^ ± 0.011	0.82 ^C^ ± 0.010	0.63 ^D^ ± 0.017	0.43 ^E^ ± 0.005	0.75 ^d^	2.13 ^A^ ± 0.10	1.98 ^A^ ± 0.01	1.84 ^A^ ± 0.07	1.71 ^AB^ ± 0.08	1.92 ^a^
*D. sissoo*	2.13 ^A^ ± 0.10	1.73 ^A^ ± 0.07	1.39 ^A^ ± 0.04	0.98 ^A^ ± 0.04	1.56 ^a^	1.37 ^AB^ ± 0.039	1.12 ^A^ ± 0.044	0.91 ^A^ ± 0.042	0.71 ^A^ ± 0.013	1.03 ^a^	1.90 ^BC^ ± 0.06 ^BC^	1.84 ^AB^ ± 0.06	1.78 ^A^ ± 0.12	1.69 ^AB^ ± 0.03	1.81 ^c^
Mean	1.91 ^a^	1.47 ^b^	1.09 ^c^	0.75 ^d^		1.17 ^a^	0.88 ^b^	0.66 ^c^	0.47 ^d^		2.00 ^a^	1.89 ^b^	1.77 ^c^	1.66 ^d^	
HSD_0.05_(Host)	0.22	0.16	0.11	0.07	0.07	0.12	0.10	0.07	0.04	0.04	0.23	0.21	0.21	0.18	0.09
HSD_0.05_(Salinity)			0.06							0.03			0.07		
HSD_0.05_(Host × Salinity)			0.17							0.10			0.24		

The values carrying different alphabetical superscripts (^A–G^) within the columns above differ significantly among themselves (*p* < 0.05). Treatment means of host and salinity compared with alphabet ^a–f^. Values are mean ± standard deviations.

**Table 4 plants-13-01162-t004:** Predicting response of biomass in tested environments through linear modeling.

Environments	Linear Models for Predicted Biomass	Coefficient of Determination (R^2^)
Control	~74.29 + 0.82 RWC + (−8.05)Pn + (−3.62) K + (−40.57) Na/K + (−90.35) MDA + 64.27APX + (−0.32) POX + 3.76 SOD + 7.75 GR + 8.48 WP + 17.57 Dia	0.98
Low salinity stress (EC_iw_~3 dS m^−1^)	~45.890 + (−8.48) Pn+ (−15.040) Chl + (−1.530) MI + (−2.920) SPAD + 26.070 Na + 13.810 K + (−81.030) MDA + 46.070 APX + 0.730 SOD + 10.710 Dia	0.98
Moderate salinity stress (EC_iw_~6 dS m^−1^)	~−26.550 +−5.42 Pn+ (−0.35) MI + (−1.83) SPAD + 6.63 Na + 8.84 K + 0.11 POX + 30.59 GR + 3.05 WP + 7.83 Dia + (−0.31) RWC	0.99
Higher salinity stress (EC_iw_~9dS m^−1^)	~−66.140 + 0.440 RWC + (−3.080) E + (−9.290) Chl + (−0.780) MI + (−1.510) SPAD + 3.380 Na + 5.660 K + 27.080 MDA + (−11.110) APX + 62.380 CAT + 15.860 GR + 13.150 Dia	0.99

**Table 5 plants-13-01162-t005:** Top-ranked host plant species preference in sandalwood.

Environments			
Control (Good Quality Irrigation Water)	Low Salinity Stress (EC_iw_~3 dS m^−1^)	Moderate Salinity Stress (EC_iw_~6 dS m^−1^)	Higher Salinity Stress (EC_iw_~9 dS m^−1^)
*M. dubia (1)*	*M. dubia (1)*	*D. sissoo (1)*	*D. sissoo (1)*
*D. sissoo (2)*	*D. sissoo (2)*	*M. dubia (2)*	*M. dubia (2)*
*C. equisetifolia (3)*	*A. indica (3)*	*A. indica (3)*	*A. indica (3)*
*A. indica (4)*	*C. equisetifolia (4)*	*C. equisetifolia (4)*	*C. equisetifolia (4)*

## Data Availability

All the data are provided in the manuscript.

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
