# Peer review of "Host Plant Modulated Physio-Biochemical Process Enhances Adaptive Response of Sandalwood (Santalum album L.) under Salinity Stress"

_plants, 2024, doi:10.3390/plants13081162_

Round 1
Reviewer 1 Report
Comments and Suggestions for Authors
My comments on the manuscript:
Major comments
The objectives are to be explained at the end of the objective in an abstract heading
Authors have to give the basic climatic data and the location (latitude and longitude) of the study area.
Authors need to explain the Experiment description in bit detail to allow others to replicate the experiments.
How could the empirical data be used to increase adaptive response of sandalwood under salinity stress? Need justification
Authors have to discuss, which parameters are influencing Sandalwood seedlings adaptation to salt stress
How the authors recommend to maintain physiological and biochemical processes of the sandalwood to respond salinity stress? Need to discuss
Please provide the study limitation and how to solve this matter
Please improve the conclusion to respond your aim of study
Minor
L 21 D. sissoo and M. dubia were appeared first time should be full scientific name.
Comments on the Quality of English Language
Minor editing of English language required
Author Response
· Reviewer Comment: The objectives are to be explained at the end of the objective in an abstract heading.
Authors Reply: Thanks for your kind suggestion. The correction made in the revised manuscript which is as follow:
Therefore, the current investigation was conducted to understand the physiological processes and antioxidant mechanisms in sandalwood along the different salinity gradients to explore the host species that could support sandalwood growth under salt affected agro-ecosystems.
· Reviewer Comment: Authors have to give the basic climatic data and the location (latitude and longitude) of the study area.
Authors Reply: The location (latitude and longitude) of the study area already provided in the sub section 4.1 Experimental area and planting material. The details of which are as follow:
The investigation was initiated in October 2020 at ICAR–CSSRI, Karnal, India (29°84'30" N and 76°85'80"E’), and terminated in May-June 2021.
Moreover, the experiment was conducted in the shade net house which has optimum growing conditions of light intensity (PAR) of 600 flux, relative humidity >60%, temperature of 25-30°C and CO2 concentration of 400 ppm. However, we have added the climate details which are as follow:
The area experiences a semi-arid and subtropical-monsoonal climate with average 350–400 mm of annual rainfall, 70–80% of which received during July to September. The coldest months are December and January, when the lowest temperature may be as low as 0 °C. The summers are extremely hot, and the mean maximum temperature varies from 40 to 45 °C during May and June months.
· Reviewer Comment: Authors need to explain the Experiment description in bit detail to allow others to replicate the experiments.
Authors Reply: The experiment details are provided in the 4.2 Section and more details about the experiment added in the revised version of the manuscript.
· Reviewer Comment: How could the empirical data be used to increase adaptive response of sandalwood under salinity stress? Need justification
Authors Reply: The empirical data regarding the host selection provide us with the potential host tree species which can be support the growth of the sandalwood in the saline conditions. Therefore, the host tree species was selected from the recommendation of the previous literature. Moreover, with the aid of morphological data, ion dynamics, physiological data and antioxidant data, the adaptive response of the sandalwood under salinity can be judged. Since, if sandalwood maintain better physiological activities, the sandalwood can be grown in the saline soils with suitable host species
· Reviewer Comment: Authors have to discuss, which parameters are influencing Sandalwood seedlings adaptation to salt stress
Authors Reply: The linear modeling indicated in Table 4 indicated the different parameters influencing the sandalwood seedling adaptation to different salinity stress levels.
· Reviewer Comment: How the authors recommend to maintain physiological and biochemical processes of the sandalwood to respond salinity stress? Need to discuss
Authors Reply: The physiological and biochemical process of the sandalwood in the control condition was compared with different salinity stress level. In this way, the suitable host species can be identified which can help to maintain the physiological and biochemical processes in the saline condition with limited membrane injury.
· Reviewer Comment: Please provide the study limitation and how to solve this matter
Authors Reply: Thanks for your kind suggestion and correction made in the revised manuscript.
· Reviewer Comment: Please improve the conclusion to respond your aim of study
Authors Reply: Compiled as per the comment.
· Reviewer Comment: L 21 D. sissoo and M. dubia were appeared first time should be full scientific name.
Authors Reply: Thanks for your kind suggestion and the correction made in the revised manuscript.
Reviewer 2 Report
Comments and Suggestions for Authors
This research has good innovation and application potential. However, the design of this paper has minor defects, and there is no control group for cultivating sandalwood. It is recommended to select a group of the most common host plants as the control group. It is suggested that the author supplement and re-analyze the corresponding data when revising the paper. The research content and level of the paper are suitable for publication in Plants. Some suggestions were put forward as follows.
1. The research progress on the mechanism of host influence on plant salt tolerance is not fully explained in the introduction.
2. Line 134: The title of Table 1 does not match its content. Photosynthetic pigment content does not belong to photosynthetic parameters.
3. Statistical analysis of differences suggests that all lowercase letters be marked2.Where is the practical application of this manuscript? It must be added.
Line 467: The object of comparative analysis should be specified in the statistical analysis method.
4. Line 149: The units of Photosynthetic rate (µmolm-2s-1) and Transpiration rate (µmolm-2s-1) are wrong. First, there is one less space after mol. Second, the unit of Transpiration rate is “mmol m-2s-1”.
5. The units of water potential and osmotic potential are not provided.
6. In Table 3, only the ion content ratio is given, and the specific content of various ions is not given. Is Mg2+ or Mg2+/Na+ ratio in the third column? The specific content of ions reflects the direct and important information of plant salt-tolerance.
7. The number of repetitions are not given in all figures or tables.
8. Check the grammar throughout the article and correct it. Proofread the article as many language errors were identified.
9. Kindly integrate the relevance or importance of study findings to the international scientific community
10. In the discussion, the mechanism of host effect on salt tolerance is not discussed enough.
I wish those changes will contribute to improve your paper.
Comments on the Quality of English LanguageMinor editing of English language required
Author Response
This research has good innovation and application potential. However, the design of this paper has minor defects, and there is no control group for cultivating sandalwood. It is recommended to select a group of the most common host plants as the control group. It is suggested that the author supplement and re-analyze the corresponding data when revising the paper. The research content and level of the paper are suitable for publication in Plants. Some suggestions were put forward as follows.
· Reviewer Comment: The research progress on the mechanism of host influence on plant salt tolerance is not fully explained in the introduction.
Authors Reply: Thanks for your kind comment. The mechanism of host influence on plant salt tolerance indicated in the introduction section as follow:
Hence, it is imperative to promptly identify appropriate host plants that can thrive in challenging conditions, like in saline soils. Particularly, salinity stress initially reduces water availability due to osmotic stress. Over time, it leads to the accumulation of detrimental ions, particularly Na+ and Cl-, which hinders growth and physico-biochemical processes, and generates reactive oxygen species (Zhao et al., 2021). Tolerant species exhibit a greater ratio of potassium (K+) to sodium (Na+) in their tissues, which enables them to grow more and produce a significant amount of biomass even when exposed to high salt levels (Verma et al. 2023).
In general, three factors— the choice of sandalwood for hosts, the ability of hosts to tolerate salinity, and regulating the physiological and biochemical processes in sandalwood—likely determine the salinity tolerance and growth of sandalwood in saline environments.
· Reviewer Comment: Line 134: The title of Table 1 does not match its content. Photosynthetic pigment content does not belong to photosynthetic parameters.
Authors Reply: Compiled as per the comment.
· Reviewer Comment: Statistical analysis of differences suggests that all lowercase letters be marked2.Where is the practical application of this manuscript? It must be added.
Authors Reply: Sorry, I did not get the comment. However, we have added the statistical difference in the figure 2. Moreover, overall, practicality of the current investigation lies in the identification of the suitable host for the control, EC 3, EC 6 and EC 6 on physiological and biological grounds. Since, it is well established that, morphological data alone does not truly represent the sandalwood behavior, while by understanding the physiological and biochemical process the actual behaviour of the sandalwood in the different salinity can be fully understand.
· Reviewer Comment: Line 467: The object of comparative analysis should be specified in the statistical analysis method.
Authors Reply: The objective of the current investigation indicated in the statistical analysis method. Now, in the manuscript both One way ANOVA and Two-way ANOVA was conducted. The experimental data for each parameter of sandalwood grown with ten selected host species was analyzed using One-way analysis of variance (ANOVA) at four salinity levels separately using STAR statistical software [75]. This analysis was conducted to determine the most suitable host species for each salinity level. Simultaneously, Two-way analysis of variance (ANOVA) was also conducted to draw the cumulative influence of the host species and salinity level on the Sandalwood.
· Reviewer Comment: Line 149: The units of Photosynthetic rate (µmolm-2s-1) and Transpiration rate (µmolm-2s-1) are wrong. First, there is one less space after mol. Second, the unit of Transpiration rate is “mmol m-2s-1”.
Authors Reply: Thanks for your kind suggestion. The correction made in the revised version.
· Reviewer Comment: The units of water potential and osmotic potential are not provided.
Authors Reply: The correction made in the revised manuscript and units i.e., MPa provided
· Reviewer Comment: In Table 3, only the ion content ratio is given, and the specific content of various ions is not given. Is Mg2+ or Mg2+/Na+ ratio in the third column? The specific content of ions reflects the direct and important information of plant salt-tolerance.
Authors Reply: Thanks, for your kind suggestion. The specific content of the ions provided in the supplementary file. Kindly refer table No. Table S3, Table S4, Table S5, and Table S6. Moreover, in third column of table 3, the Ca2+/Mg2+ ratio provided.
· Reviewer Comment: The number of repetitions are not given in all figures or tables.
Authors Reply: Thanks for suggestion. The replication detail i.e., (n=5) provided in the revised version.
· Reviewer Comment: Check the grammar throughout the article and correct it. Proofread the article as many language errors were identified.
Authors Reply: We have checked and proofread the manuscript for language errors. Thanks for your kind comments
· Reviewer Comment: Kindly integrate the relevance or importance of study findings to the international scientific community
Authors Reply: Compiled as per the comment and the suggestion incorporated in the revised manuscript.
· Reviewer Comment: In the discussion, the mechanism of host effect on salt tolerance is not discussed enough.
Authors Reply: The mechanism underlying host effect on salt tolerance provided in the introduction section as well as in Discussion section. Moreover, the mechanism for the different parameters also improved by adding recent literature.
Reviewer 3 Report
Comments and Suggestions for Authors
The authors investigated the suitable host plant for sandalwood under saline conditions and the assessment of salinity induced changes in physiology and redox homeostasis mechanism in sandalwood. This information can contribute to enhance the income and sustainable livelihood of marginal Indian farmers residing in degraded ecosystems. The research represents a study that involved a considerable volume of work and analyses. However, the manuscript has several lacks, such that I regret to recommend that the manuscript is not acceptable for publication in its present form. All detailed comments and suggestions can be found in the attached file.

Comments on the Quality of English LanguageMinor editing of English language required.
Author Response
The authors investigated the suitable host plant for sandalwood under saline conditions and the assessment of salinity induced changes in physiology and redox homeostasis mechanism in sandalwood. This information can contribute to enhance the income and sustainable livelihood of marginal Indian farmers residing in degraded ecosystems. The research represents a study that involved a considerable volume of work and analyses. However, the manuscript has several lacks, such that I regret to recommend that the manuscript is not acceptable for publication in its present form. All detailed comments and suggestions can be found in the attached file.
· Reviewer Comment: I recommend you to ad in the title the Latin name of sandalwood (Santalum album).
Authors Reply: Compiled as per the comment
· Reviewer Comment: Replace hosts with host species (in keywords)
Authors Reply: Correction made in the revised version.
· Reviewer Comment: Use the full Latin names of host species D. sissoo (Dalbergia sissoo), M. dubia (Monstera dubia) and A.indica (Azadirachta indica) in the first mentioned place (Abstract). Do the same for other host species: C. equisetifolia, P. emblica, C. aurantium, ....
Authors Reply: Compiled as per the suggestion.
· Reviewer Comment: Line 45, replace 16.84 MT with 16.84 million tonnes, since the abbreviation MT is used only once in the text
Authors Reply: Thanks for your suggestion. The correction made in the revision.
· Reviewer Comment: The text from lines 48-51 and 64-68, has no reference(s).
Authors Reply: The relevant reference added.
· Reviewer Comment: Considering your finding (lines 84-85) that “Acacia acuminata, offers some advantages in terms of improved growth of S.album”, why you didn't use this species (A.acuminata ) in the study?
Authors Reply: In the current study, saline-tolerant host plants are chosen based on existing literature. Furthermore, Acacia acuminata is unique to the Australian continent and not commonly seen on the Indian subcontinent. That is why Acacia acuminata was not considered in the current study, while Acacia ampliceps was considered as a host.
· Reviewer Comment: In Table S4 there is no information regarding the variation of diameter and biomass of sandalwood plants under different salinity stress, as you mentioned in the lines 107-113? You have to add this information (data) associated to the text from lines 107-113. Also, this text must be inserted in a subsection (2.1. Morphological parameters) of the Results section.
Authors Reply: The suggestions compiled in the revised version of the manuscript and the tables of the morphological parameters provided in Table S1.
· Reviewer Comment: In the title of Table 1, 2, 3 replace hosts with host species (as in the tables). Insert in the footer of the Table 1, 2, 3 the following text: Means with different letters are significant at p≤0.05. Values are mean±SD (or SE), for Tables 1,2.
Authors Reply: The suggestion compiled in the revised version and following statement added:
The values carrying different alphabetical superscripts (A,B,C,D.,…) within the columns above, differ significantly amongst themselves (p<0.05). Values are mean ± standard deviations.
· Reviewer Comment: In Figure 1 I think it is more appropriate (frequently used) to mark each graph with a, b, c instead of I, ii, iii. Also, insert these letters near or below each graph.
Authors Reply: Compiled as per the comment and correction made in the revised version.
· Reviewer Comment: Given that the experiment was replicated five times, for better data credibility you must to insert in the Table 3 the associated values SD/SE for each mean.
Authors Reply: The values of the standard deviation added in the table 3.
· Reviewer Comment: For a better visibility, I recommend you to enlarge the graphs from Figure 3.
Authors Reply: The figures enlarged and will be uploaded as high-quality images separately.
· Reviewer Comment: Given that your experiment was organized on two factors (10 host × 4 salinity Level × 5 Replications) it's absolutely necessary to use the HSD0.05 for host x salinity interaction in order to establish the significance of the differences between the means of each host at different salinity level. Thus, you will obtain useful information (statistically argued) regarding the effect of salinity on the various parameters for each species.
Authors Reply: As per your kind suggestion, HSD0.05 for host x salinity interaction provided in the revised manuscript. Moreover for the datasets provided in the manuscript as figures, the subsequent interaction CD provided in the supplementary file. Kindly check the table Table S2 to Table S13.
· Reviewer Comment: The graph from Figure 2 is very loaded with a lot of data, I recommend you to split these information into two separate graphs, for membrane injury and lipid peroxidation, respectively. Also, you must insert letters in order to explain the significance of differences between means.
Authors Reply: As per your kind suggestion, the information split into 2 separate figures i.e., Figure 2a and Figure 2b. Moreover, the significance of difference between means also provided in the manuscript.
· Reviewer Comment: In the M&M section, as you described RWC, it is necessary to describe synthetically the other methods used for analysis.
Authors Reply: The detailed procedure for the estimation of all the parameters provided in the supplementary file 2.
· Reviewer Comment: In the M&M section you said that “five leaves were used for SPAD reading” but the data related to these measurements cannot be found in the manuscript?
Authors Reply: Thanks for indicating the error. The SPAD statement removed and instead the carotenoids method provided.
Round 2
Reviewer 1 Report
Comments and Suggestions for Authors
The authors have addressed all my comments and suggestions, I recommend to accept it in present form
Comments on the Quality of English LanguageMinor editing of English language required
Reviewer 3 Report
Comments and Suggestions for Authors
The new version of the manuscript 2925886 is significantly improved, but a few more additions are needed.
In the M&M section you must specify that the methods used for analysis are detailed in supplementary file 2. In Tables S2…13 you must indicate (using capital letters) the significance of the differences between the means of each host at different salinity level, the same as in the Tables 1, 2, 3.
Congratulations for your work!
Comments on the Quality of English LanguageMinor editing of English language required